# ATD: Augmenting CP Tensor Decomposition by Self Supervision

[1]**Chaoqi Yang**, [2]**Cheng Qian**, [1]**Navjot Singh**, [3]**Cao Xiao**, [4,5]**M Brandon Westover**,
[1]**Edgar Solomonik**, [1]**Jimeng Sun**
[1]University of Illinois Urbana-Champaign, [2]IQVIA, [3]Relativity,
[4]Massachusetts General Hospital, [5]Harvard Medical School
[1]{chaoqiy2,navjot2,solomon2,jimeng}@illinois.edu, [2]alextoqc@gmail.com,
[3]cao.xiao@relativity.com, [4,5]mwestover@mgh.harvard.edu

## Abstract

Tensor decompositions are powerful tools for dimensionality reduction and feature interpretation of multidimensional data such as signals. Existing tensor decomposition objectives (e.g., Frobenius norm) are designed for fitting raw data under statistical assumptions, which may not align with downstream classification tasks. In practice, raw input tensors can contain irrelevant information, while data augmentation techniques may be used to smooth out class-irrelevant noise in samples. This paper addresses the above challenges by proposing augmented tensor decomposition (ATD), which effectively incorporates data augmentations and self-supervised learning (SSL) to boost downstream classification. To address the non-convexity of the new augmented objective, we develop an iterative method that enables the optimization to follow an alternating least squares (ALS) fashion. We evaluate our proposed ATD on multiple datasets. It can achieve $0.8\% \sim 2.5\%$ accuracy gain over tensor-based baselines. Also, our ATD model shows comparable or better performance (e.g., up to $15\%$ in accuracy) over self-supervised and autoencoder baselines while using less than $5\%$ of learnable parameters of these baseline models. We have released our data processing and codes in https://github.com/ycq091044/ATD.

## 1 Introduction

Extracting unsupervised features from high-dimensional data is essential in various scenarios, such as physiological signals (Cong et al., 2015), hyperspectral images (Wang et al., 2017) and fMRI (Hamdi et al., 2018). Tensor decomposition models are often used for high-order feature extraction (Sidiropoulos et al., 2017). Among these, the CANDECOMP/PARAFAC (CP) decomposition is one of the most popular models. The low-rank CP tensor decompositions (Kolda and Bader, 2009) assume that the input data is composited by a small set of components, while the reduced features are the coefficients that quantify the importance of each basis, which provides a compact representation.

Under this low-rank assumption, existing tensor decomposition objectives aim to *fit individual data samples* with statistical error measures (Hong et al., 2020; Singh et al., 2021; Yang et al., 2022), e.g., Frobenius norm or KL-divergence. Though *fitness* is an essential principle for feature reduction, common objective functions do not account for downstream tasks, e.g., classification.

Contrastive self-supervised learning (SSL) (He et al., 2020) is recently popular for unsupervised feature learning, which utilizes the class-preserving data augmentations (Dao et al., 2019) and learns an encoder that can filter out class-irrelevant information. The optimization goal is to *enforce alignments* (Chen et al., 2020; Wang and Isola, 2020), ensuring that the anchor sample is closer to the positive sample (which has the same latent class as the anchor sample) than to the negative sample (which is in a different latent class) in the embedding space. In an unsupervised setting, positive

samples are given by data augmentations, while negative samples are hard to acquire (Arora et al., 2019; He et al., 2020; Chen et al., 2020). Also, previous models are mostly built on deep neural networks, which are often black-box models with tens of thousands of learnable parameters.

This paper aims to incorporate both the *fitness* and *alignment* principles into CP tensor decomposition[1] by augmenting the common fitness objective with a new self-supervised loss. The new self-supervised loss is based on the unbiased estimation of negative samples (Chuang et al., 2020), which effectively prevents the sampling bias issue (Arora et al., 2019). The purpose of our design (i.e., introducing SSL into CP tensor decomposition) is to learn class-aware tensor decomposition results for boosting the downstream tensor sample classifications. To address the non-convex subproblems from the new objective, we formulate an iterative method, which solves least squares optimization in each step with a closed-form solution. The main contributions are summarized below.

- We propose **augmented tensor decomposition**, named ATD, which learns an unsupervised CP structure decomposition by extending the original fitness objective with a self-supervised loss on the contrastiveness of similar and dissimilar tensor samples.

- We develop an iterative method to address the non-convex subproblem from the new objective, which enables our algorithm to **follow an alternative least squares fashion**. Our algorithm has *asymptotically the same complexity* of each optimization sweep as the common CP-ALS.

- We provide **extensive evaluations on four real-world datasets** and compare to recent tensor decomposition models, autoencoder models, and self-supervised models. Our method shows better or comparable prediction performance in various downstream classifications while only requiring much fewer (e.g., less than $5\%$ of) parameters than that of deep learning baselines.

## 2 Background

**Notation.** We use plain letters for scalars, such as $x$ or $X$, boldface uppercase letters for matrices, e.g., $\mathbf{X}$, boldface lowercase letters for vectors, e.g., $\mathbf{x}$, and Euler script letters for tensors, random variables of tensors, and probability distributions, e.g., $\mathcal{X}$. Tensors are multidimensional arrays indexed by three or more indices (modes). For example, an $N$-mode tensor $\mathcal{X}$ is an $N$-dimensional array of size $I_1 \times \cdots \times I_N$, where $x_{i_1,\dots,i_N}$ is the element at the $(i_1, \cdots, i_N)$-th position. For matrix $\mathbf{X}$, the $r$-th row and column are $\mathbf{x}^{(r)}$ and $\mathbf{x}_r$ respectively, while $x_{ij}$ is for the $(i, j)$-th element. $\|\mathbf{X}\|_F$ is the Frobenius norm. For vector $\mathbf{x}$, the $r$-th element is $x_r$, and we use $\|\mathbf{x}\|_2$ to denote the vector 2-norm, $\langle \cdot, \cdot \rangle$ for the vector inner product, $\circ$ for the outer product, and $[\![\cdot]\!]$ for the Kruskal product. Indices in the paper start from 1, e.g., $\mathbf{x}_1$ is the first column of the matrix.

### 2.1 Tensor Modeling

This paper aims to learn tensor bases from unlabeled data and then use the bases to build a feature extractor for downstream classification. Without loss of generality (w.r.t. tensor order), we consider the fourth-order tensor, e.g., a collection of multi-channel Electroencephalography (EEG) signals,

$$\mathcal{T} = \left[ \mathcal{T}^{(1)}, \mathcal{T}^{(2)}, \dots, \mathcal{T}^{(N)} \right] \in \mathbb{R}^{N \times I \times J \times K},$$

where $\mathcal{T}^{(n)} \in \mathbb{R}^{I \times J \times K}$. The first dimension of $\mathcal{T}$ corresponds to data samples (e.g., one for each patient), while the other three are feature dimensions (e.g., *channel by frequency by timestamp*).

**Data Model.** To model the above tensor, previous works (Kolda and Bader, 2009) assume that

- There are a set of rank-one tensor components $\mathcal{E} = \{\mathcal{E}_1, \dots, \mathcal{E}_R\}$, which are learnable;

- The tensor data sample/slice $\mathcal{T}^{(n)}$ admits a low-rank structure and can be represented as a weighted sum of these tensor components $\mathcal{E}$, where $\mathbf{x}^{(n)}$ denotes its coefficient vector;

- On top of the low-rank structure, each data sample $\mathcal{T}^{(n)}$ also contains additional element-wise i.i.d. Gaussian noise due to real-world distortion (e.g., physical noise in signal measurements).

---

[1]Our design may work for other tensor models, such as Tucker decomposition. We leave it to future work.

In the context of downstream classifications, we further assume that each sample $\mathcal{T}^{(n)}$ is semantically associated to one of the latent classes $p \in \{1, \ldots, P\}$, and we let $\mathcal{D}_p$ be the sample distribution of class-$p$. Thus, the data sample can be formulated as, $\forall n$,

$$\mathcal{T}^{(n)} = \sum_{r=1}^{R} x_r^{(n)} \cdot \mathcal{E}_r + \epsilon^{(n)} \sim \mathcal{D}_p, \quad p \in \{1, \ldots, P\}, \tag{1}$$

where

$$\mathcal{E}_r = \mathbf{a}_r \circ \mathbf{b}_r \circ \mathbf{c}_r, \quad r \in \{1, \ldots, R\},$$

$$\epsilon^{(n)} \sim_{\text{i.i.d.}} \mathcal{N}(0, \sigma), \quad \text{where } \sigma \text{ is generally small.}$$

Here $\mathbf{a}_r, \mathbf{b}_r, \mathbf{c}_r$ are the learnable parameters, which produces the rank-one component $\mathcal{E}_r$, and they are the column vectors of $\{\mathbf{A} \in \mathbb{R}^{I \times R}, \mathbf{B} \in \mathbb{R}^{J \times R}, \mathbf{C} \in \mathbb{R}^{K \times R}\}$, referred as "bases".

**CANDECOMP/PARAFAC Decomposition (CPD).** Given the above tensor model, standard CPD only captures the i.i.d. Gaussian noise by minimizing the negative log-likelihood (NLL), which results in the following standard fitness/reconstruction loss,

$$\mathcal{L}_{cpd} = \sum_{n=1}^{N} \left\| \mathcal{T}^{(n)} - [\![\mathbf{x}^{(n)}, \mathbf{A}, \mathbf{B}, \mathbf{C}]\!] \right\|_F^2 = \|\mathcal{T} - [\![\mathbf{X}, \mathbf{A}, \mathbf{B}, \mathbf{C}]\!]\|_F^2.$$

Here, the Kruskal product $[\![\cdot]\!]$ outputs a fourth-order reconstructed tensor from four input factor matrices. For consistency, if the first input is a vector, the output is considered as a third-order tensor.

## 2.2 Problem Formulation

CP decomposition seeks a low-rank reconstruction, without special consideration for the downstream task. In this paper, we are motivated to improve the CPD model by exploiting the latent classes (in an unsupervised way) and learn good bases to provide better class-aware features for classification.

**What are Good Bases?** This paper considers two design principles for good bases. The first principle is *fitness*, which requires a low-rank tensor reconstruction with the bases. Second, data samples associated with the same latent class should be decomposed into similar coefficient vectors, with the bases, while the vectors should be dissimilar if the samples are from different latent classes. This principle is called *alignment*, which is important for classification but not considered in the standard tensor decomposition. In this paper, we assess the quality of the learned bases by the performance of downstream classification, where the coefficient vectors (obtained using the bases via decomposition) are the feature inputs (into the downstream classifier).

**Working Pipelines.** To put it succinctly, the paper tackles an unsupervised learning problem while using downstream supervised classification for evaluation. The procedures are briefly outlined:

- First, we **learn** the bases $\{\mathbf{A}, \mathbf{B}, \mathbf{C}\}$ from a large set of unlabeled data. The loss function is developed in consideration of the *fitness* and *alignment* (defined in the next section) principles.

- Then, we **construct** the following feature extractor given $\{\mathbf{A}, \mathbf{B}, \mathbf{C}\}$. The feature vector of a new sample is obtained by the closed-form solution of the least squares problem ($\alpha > 0$ is a hyperparameter),

$$\mathbf{f}(\mathcal{T}^{(new)}; \mathbf{A}, \mathbf{B}, \mathbf{C}) = \arg\min_{\mathbf{x} \in \mathbb{R}^{1 \times R}} \left( \left\| \mathcal{T}^{(new)} - [\![\mathbf{x}, \mathbf{A}, \mathbf{B}, \mathbf{C}]\!] \right\|_F^2 + \alpha \|\mathbf{x}\|_2^2 \right). \tag{2}$$

  Note that, when $\mathbf{f}(\cdot)$ is applied to a batch of samples, it outputs a coefficient matrix.

- Next, we **evaluate** the feature extractor with a set of labeled data. Given $\mathbf{f}(\cdot)$, we first apply it on the labeled data to extract their features and then train an additional logistic regression model (as the downstream classifier) on top of the extracted features, so that the result of classifications will implicitly reflect how good the bases are.

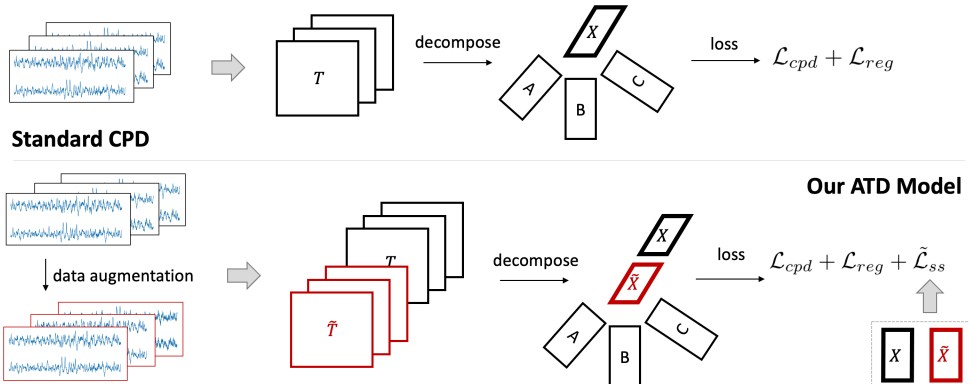

Figure 1: Standard CPD vs Our ATD Model

# 3 Augmented Tensor Decomposition

We show our model in Figure 1. The design is inspired by the recent popularity of SSL. To exploit the latent class information, we introduce class-preserving data augmentation into CPD model and design self-supervised loss to constrain the learned low-rank features (i.e., the coefficient vectors).

**Data Augmentation**[2] In general, data augmentation methods are chosen to perturb the raw data while preserving class label information (He et al., 2020; Chen et al., 2020). Given a sample $\mathcal{T}^{(n)}$, we assume that after data augmentation, its perturbation $\widetilde{\mathcal{T}}^{(n)}$ preserves the label and admits the component-based representation as in Eqn. (1).

## 3.1 Self-supervised Loss

The design of our self supervised loss corresponds to the *alignment* principle, which is based on pairwise feature similarity and dissimilarity. We call a pair of data samples from the same latent class as *positive pair*, a pair of samples from different latent classes as *negative pair* and a pair of independent samples (two random samples from the dataset) as *random pair*. Intuitively, an anchor plus a positive sample composes a positive pair, similarly for negative pairs and random pairs. In this work, our self-supervised loss aims to maximize the feature similarity between positive pairs and minimizes that between negative pairs *in an unsupervised way* (no labels during optimization).

Formally, let $\mathcal{X}_p, \mathcal{Y}_p$ be discrete random variables (of tensor samples) distributed as $\mathcal{D}_p$, $p \in \{1, \ldots, P\}$. We want to minimize the following objective when *no class labels are given*,

$$\mathcal{L}_{ss} = \mathcal{L}_{pos} + \lambda \mathcal{L}_{neg},$$

where $\lambda \geq 1$ is a hyperparameter and

$$\mathcal{L}_{pos} = -\mathbb{E}\left[\mathrm{sim}\left(\mathbf{f}\left(\mathcal{X}_p\right), \mathbf{f}\left(\mathcal{Y}_q\right)\right) \mid p = q\right],$$
$$\mathcal{L}_{neg} = \mathbb{E}\left[\mathrm{sim}\left(\mathbf{f}\left(\mathcal{X}_p\right), \mathbf{f}\left(\mathcal{Y}_q\right)\right) \mid p \neq q\right]. \tag{3}$$

Here $\mathcal{L}_{pos}$ maximizes the feature similarity of positive pairs while $\mathcal{L}_{neg}$ minimizes the feature similarity of negative pairs. $\mathbf{f}(\cdot)$ is the feature extractor, defined in Eqn. (2), and the similarity measure is given by cosine distance, parameterized by two random variables,

$$\mathrm{sim}\left(\mathbf{f}\left(\mathcal{X}_p\right), \mathbf{f}\left(\mathcal{Y}_q\right)\right) = \left\langle \frac{\mathbf{f}(\mathcal{X}_p)}{\|\mathbf{f}\left(\mathcal{X}_p\right)\|_2}, \frac{\mathbf{f}(\mathcal{Y}_q)}{\|\mathbf{f}\left(\mathcal{Y}_q\right)\|_2} \right\rangle.$$

To this end, the key is to implement the self-supervised loss, i.e., Eqn. (3), in an unsupervised setting. Specifically, we want to construct the sampler of positive pairs and the sampler of negative pairs with unlabeled data only. The sampler of positive pairs (in short, positive sampler) can be easily approximated by data augmentation techniques, which provides "surrogate" positive pairs (given any

---

[2]We provide further discussions and ablation studies on data augmentation in Appendix C.3 and C.5.

sample as the anchor, we apply data augmentation methods to generate a perturbed data as positive sample, and then the anchor plus the perturbed data is a positive pair). However, the negative sampler is infeasible, without labels. As a compromise, previous works (He et al., 2020; Chen et al., 2020) consider using random sampler to replace the negative sampler given that the random sampler can be easily achieved by picking two independent samples from the dataset. However, this practice is shown to induce sampling bias and hurts the learned representation (Arora et al., 2019; Chuang et al., 2020) since a random pair may be from the same latent class.

**Construction of Negative Sampler.** In this paper, we consider using the *law of total probability* to construct the negative sampler in a statistical way. Formally, assume $r_p$ is the label rate of latent class-$p$ (thus, we have $\sum_p r_p = 1$), we apply the law of total probability and the following holds,

$$
\begin{aligned}
\mathbb{E}\left[\text{sim}\left(\mathbf{f}\left(\mathcal{X}_p\right), \mathbf{f}\left(\mathcal{Y}_q\right)\right) \mid p \neq q\right] &= \sum_{p=1}^{P} r_p \sum_{q \neq p} \frac{r_q}{1-r_p} \mathbb{E}_{p,q}\left[\text{sim}\left(\mathbf{f}\left(\mathcal{X}_p\right), \mathbf{f}\left(\mathcal{Y}_q\right)\right)\right] \\
&= -\sum_{p=1}^{P} \frac{r_p r_p}{1-r_p} \mathbb{E}_{p,q}\left[\text{sim}\left(\mathbf{f}\left(\mathcal{X}_p\right), \mathbf{f}\left(\mathcal{Y}_q\right)\right) \mid p = q\right] + \sum_{p=1}^{P}\sum_{q=1}^{P} \frac{r_p r_q}{1-r_p} \mathbb{E}_{p,q}\left[\text{sim}\left(\mathbf{f}\left(\mathcal{X}_p\right), \mathbf{f}\left(\mathcal{Y}_q\right)\right)\right] \\
&= -\mathbb{E}\left[\frac{r_p}{1-r_p}\text{sim}\left(\mathbf{f}\left(\mathcal{X}_p\right), \mathbf{f}\left(\mathcal{Y}_q\right)\right) \mid p = q\right] + \mathbb{E}\left[\frac{1}{1-r_p}\text{sim}\left(\mathbf{f}\left(\mathcal{X}_q\right), \mathbf{f}\left(\mathcal{Y}_q\right)\right)\right].
\end{aligned}
\tag{4}
$$

Here, the usage of $\mathbb{E}[\cdot]$ means that the expectation is taken over four interdependent random variables, i.e., $p, q, \mathcal{X}_p, \mathcal{Y}_q$, while $\mathbb{E}_{p,q}[\cdot]$ means that $p, q$ is fixed and thus it is only taken over two random variables, i.e., $\mathcal{X}_p, \mathcal{Y}_q$. The result shows that the negative sampler can be equivalently replaced by a weighted combination of the random sampler and positive sampler. Here we do not have access to $r_p, \forall p$ with unlabeled data, this issue is dealt with later.

**Self-supervised Loss.** Consequently, we can reformulate our self-supervised loss as,

$$
\begin{aligned}
\mathcal{L}_{ss} &= \mathcal{L}_{pos} + \lambda \mathcal{L}_{neg} \\
&= -\mathbb{E}\left[\text{sim}\left(\mathbf{f}\left(\mathcal{X}_p\right), \mathbf{f}\left(\mathcal{Y}_q\right)\right) \mid p = q\right] + \lambda \mathbb{E}\left[\text{sim}\left(\mathbf{f}\left(\mathcal{X}_p\right), \mathbf{f}\left(\mathcal{Y}_q\right)\right) \mid p \neq q\right] \tag{5} \\
&= \mathbb{E}\left[\frac{\lambda}{1-r_p}\text{sim}\left(\mathbf{f}\left(\mathcal{X}_p\right), \mathbf{f}\left(\mathcal{Y}_q\right)\right)\right] - \mathbb{E}\left[\left(\frac{\lambda r_p}{1-r_p} + 1\right)\text{sim}\left(\mathbf{f}\left(\mathcal{X}_p\right), \mathbf{f}\left(\mathcal{Y}_q\right)\right) \mid p = q\right]. \tag{6}
\end{aligned}
$$

From Eqn. (5) to Eqn. (6), we use the results in Eqn. (4)

**Two-sided Bound.** The above form still requires label rate information, i.e., $r_p, \forall p$, we therefore consider using the following approximation to the above loss $\mathcal{L}_{ss}$,

$$
\mathcal{L}_{ss}^{\Theta}(\gamma) = (\gamma + 1)\mathbb{E}\left[\text{sim}\left(\mathbf{f}\left(\mathcal{X}_p\right), \mathbf{f}\left(\mathcal{Y}_q\right)\right)\right] - \mathbb{E}\left[\text{sim}\left(\mathbf{f}\left(\mathcal{X}_p\right), \mathbf{f}\left(\mathcal{Y}_q\right)\right) \mid p = q\right]. \tag{7}
$$

Here, $\gamma \geq 0$ is a hyperparameter, while $\mathcal{L}_{ss}$ can be bounded as (derivations in appendix E),

$$
C_1 \mathcal{L}_{ss}^{\Theta}\left(\frac{\lambda - 1}{C_1}\right) \leq \mathcal{L}_{ss} \leq C_2 \mathcal{L}_{ss}^{\Theta}\left(\frac{\lambda - 1}{C_2}\right), \quad C_1 = 1 + \max_p \frac{\lambda r_p}{1-r_p}, C_2 = 1 + \min_p \frac{\lambda r_p}{1-r_p}. \tag{8}
$$

The equivalence is established when $C_1 = C_2$, i.e., the class labels are balanced. To simplify the derivation, we ignore $\lambda$ in the following and let $\gamma$ be a new hyperparameter. Also, the constants $C_1$ and $C_2$ can be absorbed into a weight hyperparameter $\beta$, given in the next section. This bound implies that, an easy-to-compute $\beta \mathcal{L}_{ss}^{\Theta}(\gamma)$ is often a good approximation of $\mathcal{L}_{ss}$ for some $\beta$. The next section specifies how to compute $\beta \mathcal{L}_{ss}^{\Theta}(\gamma)$ unsupervisedly as our *empirical self-supervised loss*.

### 3.2 The Objective of ATD

**Empirical Estimator.** We obtain an empirical estimator of $\mathcal{L}_{ss}^{\Theta}$ with Monte Carlo method. Suppose $\mathcal{T}$ and $\tilde{\mathcal{T}}$ are the input tensor and the augmented tensor respectively, and $\mathbf{X} = \mathbf{f}(\mathcal{T}), \tilde{\mathbf{X}} = \mathbf{f}(\tilde{\mathcal{T}}) \in \mathbb{R}^{N \times R}$ are the coefficient/feature matrices. We use the row vectors of $\mathbf{X}, \tilde{\mathbf{X}}$ to estimate Eqn. (7).

The first term $\mathbb{E}\left[\text{sim}\left(\mathbf{f}\left(\mathcal{X}_p\right), \mathbf{f}\left(\mathcal{Y}_q\right)\right)\right]$ essentially requires a random sampler, which is approximated by the average cosine similarity of all possible pairs of non-corresponding row vectors of $\mathbf{X}, \tilde{\mathbf{X}}$, while

the second term $\mathbb{E}\left[\operatorname{sim}\left(\mathbf{f}\left(\mathcal{X}_p\right), \mathbf{f}\left(\mathcal{Y}_q\right)\right) \mid p=q\right]$ requires a positive sampler, which is estimated by the average cosine similarity of all pairs of corresponding row vectors,

$$
\begin{aligned}
\tilde{\mathcal{L}}_{ss}^{\Theta}(\gamma) &= (\gamma+1) \cdot \frac{1}{N(N-1)} \sum_{n=1}^{N} \sum_{s \neq n}^{N}\left\langle\frac{\mathbf{x}^{(n)}}{\|\mathbf{x}^{(n)}\|_2}, \frac{\tilde{\mathbf{x}}^{(s)}}{\|\tilde{\mathbf{x}}^{(s)}\|_2}\right\rangle - \frac{1}{N} \sum_{n=1}^{N}\left\langle\frac{\mathbf{x}^{(n)}}{\|\mathbf{x}^{(n)}\|_2}, \frac{\tilde{\mathbf{x}}^{(n)}}{\|\tilde{\mathbf{x}}^{(n)}\|_2}\right\rangle \\
&= \operatorname{Tr}\left(\mathbf{X}^{\top} \mathbf{D}(\mathbf{X}) \mathbf{G}(\gamma) \mathbf{D}(\tilde{\mathbf{X}}) \tilde{\mathbf{X}}\right),
\end{aligned}
\tag{9}
$$

where $\mathbf{D}(\mathbf{X})=\operatorname{diag}\left(\frac{1}{\|\mathbf{x}^{(1)}\|_2}, \cdots, \frac{1}{\|\mathbf{x}^{(N)}\|_2}\right)$ is the row-wise scaling matrix and

$$
\mathbf{G}(\gamma)=\begin{bmatrix}
-\frac{1}{N} & \frac{\gamma+1}{N(N-1)} & \cdots & \frac{\gamma+1}{N(N-1)} \\
\frac{\gamma+1}{N(N-1)} & -\frac{1}{N} & \cdots & \frac{\gamma+1}{N(N-1)} \\
\cdots & \cdots & \cdots & \cdots \\
\frac{\gamma+1}{N(N-1)} & \frac{\gamma+1}{N(N-1)} & \cdots & -\frac{1}{N}
\end{bmatrix}.
$$

Note that, the form in Eqn. (9) is significantly different from tensor discriminant analysis (Jia et al., 2014; Tao et al., 2007), which integrates the actual label information as a regularizer and is also different from graph regularized tensor decomposition (Cai et al., 2010), which incorporates side information, such as geometrical positions (Maki et al., 2018). Compared to standard noise contrastive estimation (NCE) (Gutmann and Hyvärinen, 2010; Chen et al., 2020) in the area of contrastive SSL, our SSL form in Eqn. (9) considers a subtraction form instead of the softmax formulation, making it amenable to quadratic optimization, as we will show in Sec. 3.3.

**Overall Objective.** According to Eqn. (8), the self supervised loss $\mathcal{L}_{ss}$ is bounded by $\mathcal{L}_{ss}^{\Theta}(\gamma)$, while the constants (i.e., $C_1, C_2$) can be absorbed into a weight hyperparameter $\beta$. We let the empirical self-supervised loss, $\tilde{\mathcal{L}}_{ss}=\beta \tilde{\mathcal{L}}_{ss}^{\Theta}(\gamma)$. Our objective follows both the *fitness* (i.e., CPD reconstruction loss) and *alignment* (i.e., self-supervised loss) principles, while also considering Tikhonov regularization (Golub and Von Matt, 1997) to constrain the scale of all parameters,

$$
\mathcal{L}=\mathcal{L}_{cpd}+\mathcal{L}_{reg}+\tilde{\mathcal{L}}_{ss},
\tag{10}
$$

where

$$
\begin{aligned}
\mathcal{L}_{cpd} &= \|\mathcal{T}-[\![\mathbf{X}, \mathbf{A}, \mathbf{B}, \mathbf{C}]\!]\|_F^2 + \left\|\tilde{\mathcal{T}}-[\![\tilde{\mathbf{X}}, \mathbf{A}, \mathbf{B}, \mathbf{C}]\!]\right\|_F^2, \\
\mathcal{L}_{reg} &= \alpha\left(\|\mathbf{X}\|_F^2 + \|\tilde{\mathbf{X}}\|_F^2 + \|\mathbf{A}\|_F^2 + \|\mathbf{B}\|_F^2 + \|\mathbf{C}\|_F^2\right), \\
\tilde{\mathcal{L}}_{ss} &= \beta \tilde{\mathcal{L}}_{ss}^{\Theta}(\gamma)=\beta \operatorname{Tr}\left(\mathbf{X}^{\top} \mathbf{D}(\mathbf{X}) \mathbf{G}(\gamma) \mathbf{D}(\tilde{\mathbf{X}}) \tilde{\mathbf{X}}\right).
\end{aligned}
\tag{11}
$$

The objective has (i) three hyperparameters, $\gamma, \alpha, \beta > 0$; (ii) five factor matrices, $\mathbf{A}, \mathbf{B}, \mathbf{C}, \mathbf{X}, \tilde{\mathbf{X}}$.

### 3.3 Alternating Least Squares Optimization

Ideally, we would like to use alternative least squares (ALS) algorithm (Sidiropoulos et al., 2017) for optimizing the objective, where each factor matrix is updated in a sequence by solving least squares subproblems. With large scale tensors, we can also resort to mini-batch stochastic ALS (Cao et al., 2020) to reduce memory footprint of the decomposition. However, the objective in Eqn. (11) is non-convex to $\mathbf{X}$ and $\tilde{\mathbf{X}}$, respectively. For addressing the non-convex subproblem, we design an iterative method in this section, which only involves solving least square problems.

**Addressing Non-convex Subproblem.** We use the subproblem of $\mathbf{X}$ as an example. Given $\mathbf{A}$, $\mathbf{B}$, $\mathbf{C}$, $\tilde{\mathbf{X}}$ fixed, we want to minimize Eqn. (10) by finding the optimal solution, denoted as $\mathbf{X}^*$,

$$
\mathbf{X}^* \leftarrow \underset{\mathbf{X}}{\arg\min}\left(\|\mathcal{T}-[\![\mathbf{X}, \mathbf{A}, \mathbf{B}, \mathbf{C}]\!]\|_F^2 + \alpha\|\mathbf{X}\|_F^2 + \beta \operatorname{Tr}\left(\mathbf{X}^{\top} \mathbf{D}(\mathbf{X}) \mathbf{G}(\gamma) \mathbf{D}(\tilde{\mathbf{X}}) \tilde{\mathbf{X}}\right)\right).
\tag{12}
$$

- First, we are interested to find that the matrix-formed problem in Eqn. (12) can be decomposed into row-wise subproblems. To obtain the solution of Eqn. (12), it is suffice to solve each subproblem

independently. Let us consider the subproblem of the $k$-th row, which is

$$\arg\min_{\mathbf{x}} \left( \left\| \mathcal{T}^{(k)} - [\![\mathbf{x}, \mathbf{A}, \mathbf{B}, \mathbf{C}]\!] \right\|_F^2 + \alpha\|\mathbf{x}\|_F^2 + \beta\mathrm{Tr}\left( \frac{\mathbf{x}^\top}{\|\mathbf{x}\|_2} \mathbf{g}^{(k)}\mathbf{D}(\tilde{\mathbf{X}})\tilde{\mathbf{X}} \right) \right), \qquad (13)$$

where $\mathcal{T}^{(k)}$ is the $k$-th slice of $\mathcal{T}$, and $\mathbf{g}^{(k)}$ is the $k$-th row of $\mathbf{G}(\gamma)$.

- Here, Eqn. (13) is still non-convex with respect to $\mathbf{x}$. We let the derivative of Eqn. (13) objective to be zero and arrange the terms, which yields,

$$\mathbf{x} = \mathbf{v}_1\mathbf{V}_3 - \frac{\beta\mathbf{v}_2}{2\|\mathbf{x}\|_2}\left( \mathbf{I} - \frac{\mathbf{x}^\top\mathbf{x}}{\|\mathbf{x}\|_2^2} \right)\mathbf{V}_3, \qquad (14)$$

where

$$\mathbf{v}_1 = \mathbf{T}_1^{(k)}(\mathbf{A}\odot\mathbf{B}\odot\mathbf{C}), \quad \mathbf{v}_2 = \mathbf{g}^{(k)}\mathbf{D}(\tilde{\mathbf{X}})\tilde{\mathbf{X}}, \quad \mathbf{V}_3 = \left(\mathbf{A}^\top\mathbf{A}*\mathbf{B}^\top\mathbf{B}*\mathbf{C}^\top\mathbf{C}+\alpha\mathbf{I}\right)^{-1}.$$

Here $\mathbf{x}, \mathbf{v}_1, \mathbf{v}_2$ are row vectors and $\mathbf{V}_3$ is a matrix. $\mathbf{T}_1^{(k)}$ is the 1-mode unfolding of $\mathcal{T}^{(k)}$, $\odot$ is the Khatri-Rao product and $*$ is the Hadamard product (i.e., element-wise product).

- We consider the following iterative rule and the fixed point is a solution for Eqn. (14), which is a stationary point of Eqn. (13),

$$\mathbf{x}_{\mathrm{impr}} = \mathbf{v}_1\mathbf{V}_3 - \frac{\beta\mathbf{v}_2}{2\|\mathbf{x}_{\mathrm{init}}\|_2}\left( \mathbf{I} - \frac{\mathbf{x}_{\mathrm{init}}^\top\mathbf{x}_{\mathrm{init}}}{\|\mathbf{x}_{\mathrm{init}}\|_2^2} \right)\mathbf{V}_3. \qquad (15)$$

We use an initial guess $\mathbf{x}_{\mathrm{init}}$ (obtained by solving Eqn. (13) with while $\beta = 0$, which is a least squares problem) to start. Then, we repeat Eqn. (15) for each row (i.e., each $k$) independently and let the improved guess be the initial guess, $\mathbf{x}_{\mathrm{init}} \leftarrow \mathbf{x}_{\mathrm{impr}}$, to iteratively improve the result.

Theorem 1 (proof in Appendix A) ensures that Eqn. (15) converges linearly if $\beta$ is chosen to be sufficiently small. In Appendix B, we verify the liner convergence and also empirically show that one round of Eqn. (15) is sufficient in our experiment, where $\beta = 2$. The non-convex subproblem of $\tilde{\mathbf{X}}$ can be solved in the same way.

**Theorem 1.** *Given non-zero row vectors* $\mathbf{v}_1, \mathbf{v}_2, \mathbf{u}^0 \in \mathbb{R}^d$, *non-singular matrix* $\mathbf{V}_3 \in \mathbb{R}^{d\times d}$ *and* $\beta > 0$. *The sequence* $\{\mathbf{u}^t\}$, *generated by* $\mathbf{u}^{t+1} = \mathbf{v}_1\mathbf{V}_3 - \frac{\beta\mathbf{v}_2}{2\|\mathbf{u}^t\|_2}\left(\mathbf{I} - \frac{\mathbf{u}^{t\top}\mathbf{u}^t}{\|\mathbf{u}^t\|_2^2}\right)\mathbf{V}_3$, *satisfies,*

$$\left\|\mathbf{u}^{t+1} - \mathbf{u}^*\right\|_2 \leq \frac{\beta(2m+M)\|\mathbf{v}_2\|_2\|\mathbf{V}_3\|_F}{m^3}\left\|\mathbf{u}^t - \mathbf{u}^*\right\|_2,$$

*where* $\mathbf{u}^*$ *is the fixed point and* $m = \min_t\|\mathbf{u}^t\|_2$, $M = \max_t\|\mathbf{u}^t\|_2$ *are the bound of the sequence. With a good* $\mathbf{u}^0$ *and a sufficiently small* $\beta$, *we can safely assume* $0 < m \leq M < \infty$.

**Optimization Procedures.** To minimize Eqn. (10), we alternatively update $\mathbf{A}, \mathbf{B}, \mathbf{C}, \mathbf{X}$, and $\tilde{\mathbf{X}}$, where each subproblem involves only solving least squares problems with closed-form solutions (summarized in Algorithm 1). With large-scale tensors (as in the experiments), we optimizes the factors in mini-batches. Between mini-batches, the basis factors $\mathbf{A}, \mathbf{B}, \mathbf{C}$ are shared and updated incrementally. We show the batch algorithm in Appendix D. The computation head of the algorithm is matricized tensor times Khatri-Rao product (MTTKRP). The complexity of our optimization algorithm is asymptotically the same as applying CP-ALS on the original tensor $\mathcal{T}$ with the same rank $R$, which costs $O(NIJKR)$ to *sweep* over the whole tensor once.

## 4 Experiments

This section presents the experimental evaluations. Due to space limitation, additional details, including data augmentations and baseline implementation, are presented in Appendix C.

### 4.1 Experimental Setup

**Data Preparation.** We use four real-world datasets: (i) *Sleep-EDF* (Kemp et al., 2000), which contains EOG, EMG and EEG Polysomnography recordings; (ii) human activity recognition *(HAR)*

---

**Algorithm 1:** Alternating Least Squares

---

1   **Input:** Data tensor $\mathcal{T} \in \mathbb{R}^{N \times I \times J \times K}$; initialized $\mathbf{A}, \mathbf{B}, \mathbf{C}, \tilde{\mathbf{X}}, \mathbf{X}$; other hyperparameters $\alpha, \beta, \gamma$;

2   Obtain the augmented tensor $\tilde{\mathcal{T}}$;

3   **repeat**

4     Use $\mathbf{A}, \mathbf{B}, \mathbf{C}, \tilde{\mathbf{X}}$ to update $\mathbf{X}$ by our iterative rules (one iteration) in Eqn. (15);

5     Use $\mathbf{A}, \mathbf{B}, \mathbf{C}, \mathbf{X}$ to update $\tilde{\mathbf{X}}$ by our iterative rules (one iteration) in Eqn. (15);

6     Use $\mathbf{B}, \mathbf{C}, \mathbf{X}, \tilde{\mathbf{X}}$ to update $\mathbf{A}$ by solving least squares problem;

7     Use $\mathbf{A}, \mathbf{C}, \mathbf{X}, \tilde{\mathbf{X}}$ to update $\mathbf{B}$ by solving least squares problem;

8     Use $\mathbf{A}, \mathbf{B}, \mathbf{X}, \tilde{\mathbf{X}}$ to update $\mathbf{C}$ by solving least squares problem;

9   **until** *max sweep exceeds or change of loss $< 0.1\%$ within 3 consecutive sweeps*;

10   **Output:** the learned bases $\{\mathbf{A}, \mathbf{B}, \mathbf{C}\}$.

---

Table 1: Dataset Statistics

| Name | Data Sample Format | Augmentations | # Unlabeled ($N$) | # Training | # Test | Task | # Class |
|---|---|---|---|---|---|---|---|
| Sleep-EDF | $I \times J \times K$: $14 \times 129 \times 86$ | (a), (b) | 331,208 | 42,803 | 41,078 | Sleep Staging | 5 |
| HAR | $I \times J \times K$: $18 \times 33 \times 33$ | (a), (b), (c) | 7,352 | 1,473 | 1,474 | Activity Recognition | 6 |
| PTB-XL | $I \times J \times K$: $24 \times 129 \times 75$ | (a), (b) | 17,469 | 2,183 | 2,185 | Gender Identification | 2 |
| MGH | $I \times J \times K$: $12 \times 257 \times 43$ | (a), (b) | 4,377,170 | 238,312 | 248,041 | Sleep Staging | 5 |

\* We report the data format after short time Fourier transform (STFT) in Appendix C.2, which is used to obtain the spectrogram.

(Anguita et al., 2013) with smartphone accelerometer and gyroscope data; (iii) Physikalisch Technische Bundesanstalt large scale cardiology database *(PTB-XL)* (Alday et al., 2020) with 12-lead ECG signals; (iv) Massachusetts General Hospital *(MGH)* (Biswal et al., 2018) datasets with multi-channel EEG waves. All datasets are split into three disjoint sets (i.e., unlabeled, training and test) by subjects, while training and test sets have labels. Basic statistics are shown in Table 1. All models (baselines and our `ATD`) use the same augmentation techniques: (a) jittering, (b) bandpass filtering, and (c) 3D position rotation. We provide an ablation study on the augmentation methods in Appendix C.5.

**Baseline Methods.**   We include the following comparison models from different perspectives:

- **Tensor based models**: `ATD`$_{ss-}$ is our variant, which removes the self-supervised loss from the objective in Eqn. (10); *Stochastic alternating least squares (SALS)* applies on the the CPD objective with Tikhonov regularizer, which works on large tensors; *Graph regularized SALS (GR-SALS)* augments the objective of SALS with a graph regularizer (Maki et al., 2018; Cai et al., 2010), define as $\text{Tr}\left(\mathbf{X}^\top \mathbf{G} \mathbf{X}\right)$.

- **Self-supervised models**: *SimCLR-$r$* (Chen et al., 2020) and *BYOL-$r$* (Grill et al., 2020) are two popular SSL models with their own objective functions, where $r$ indicates the size of the output representation.

- **Auto-encoder models**: *AE-$r$* denotes a CNN based autoencoder with mean square error (MSE) reconstruction loss, and *AE$_{ss}$-$r$* denotes the same autoencoder model with standard NCE loss in the bottleneck layer, where $r$ is the representation size.

All models use the unlabeled set to train a feature encoder and use training and test sets to evaluate. Note that, deep neural network models use the same CNN backbone. In Appendix C.8, we have also compared with two recent supervised tensor learning models, which shows the usefulness of our `ATD` and the large unlabeled set, especially in low-label rate scenarios.

**Evaluation and Environments.**   We evaluate model performance mainly based on *classification accuracy*, where we train an additional logistic classifier (He et al., 2020) on top of the feature encoder. Also, for different models, we compare their *number of learnable parameters*. The experiments are implemented by *Python 3.8.5, Torch 1.8.0+cu111* on a Linux workstation with 256 GB memory, 32 core CPUs (3.70 GHz, 128 MB cache), two RTX 3090 GPUs (24 GB memory each). All training is performed on the GPU. For tensor based models, we use $R = 32$ and implement the pipeline in CUDA manually, instead of using *torch-autograd*.

### 4.2 Experimental Results

This section shows the experimental results on downstream classification. We use all the unlabeled data to train the encoder or feature extractor, and use training data (since Sleep-EDF and MGH

Table 2: Result of Downstream Classification (%). The table shows that our `ATD` can provide comparable or better performance over all baselines with fewer parameters, especially deep learning models. It also shows the usefulness of considering both *fitness* and *alignment* as part of the objective.

| | Sleep-EDF (5,000) | | HAR (1,473) | | PTB-XL (2,183) | | MGH (5,000) | |
|---|---|---|---|---|---|---|---|---|
| | Accuracy | # of Params. | Accuracy | # of Params. | Accuracy | # of Params. | Accuracy | # of Params. |
| **Self-sup models:** | | | | | | | | |
| SimCLR-32 | $84.98 \pm 0.358$ | 210,384 | $74.75 \pm 0.723$ | 53,286 | $69.25 \pm 0.355$ | 200,960 | $67.34 \pm 0.970$ | 212,624 |
| SimCLR-128 | $\mathbf{85.19 \pm 0.358}$ | 222,768 | $76.69 \pm 0.697$ | 65,670 | $68.19 \pm 0.793$ | 237,920 | $66.98 \pm 1.331$ | 246,608 |
| BYOL-32 | $84.29 \pm 0.405$ | 211,440 | $73.71 \pm 2.832$ | 54,342 | $65.08 \pm 1.535$ | 202,016 | $68.83 \pm 1.168$ | 214,736 |
| BYOL-128 | $83.26 \pm 0.337$ | 239,280 | $71.79 \pm 1.866$ | 82,182 | $65.49 \pm 0.612$ | 254,432 | $68.55 \pm 1.339$ | 279,632 |
| **Auto-encoders:** | | | | | | | | |
| AE-32 | $74.78 \pm 0.723$ | 217,216 | $63.13 \pm 0.775$ | 62,940 | $59.01 \pm 0.896$ | 224,528 | $68.58 \pm 0.427$ | 220,088 |
| AE-128 | $75.17 \pm 0.897$ | 241,888 | $60.52 \pm 1.604$ | 87,612 | $58.29 \pm 0.412$ | 298,352 | $67.05 \pm 1.375$ | 257,048 |
| $AE_{ss}$-32 | $80.92 \pm 0.345$ | 217,216 | $71.70 \pm 2.135$ | 62,940 | $68.47 \pm 0.231$ | 224,528 | $71.46 \pm 0.386$ | 220,088 |
| $AE_{ss}$-128 | $81.84 \pm 0.259$ | 241,888 | $72.43 \pm 1.370$ | 87,612 | $68.88 \pm 0.604$ | 298,352 | $70.19 \pm 0.617$ | 257,048 |
| **Tensor models:** | | | | | | | | |
| SALS | $84.27 \pm 0.481$ | 7,328 | $91.86 \pm 0.295$ | 2,688 | $69.15 \pm 0.483$ | 7,296 | $71.93 \pm 0.379$ | 9,984 |
| GR-SALS | $84.33 \pm 0.356$ | 7,328 | $92.33 \pm 0.282$ | 2,688 | $69.02 \pm 0.477$ | 7,296 | $72.35 \pm 0.228$ | 9,984 |
| $ATD_{ss-}$ | $84.19 \pm 0.221$ | 7,328 | $92.41 \pm 0.391$ | 2,688 | $69.38 \pm 0.612$ | 7,296 | $72.78 \pm 0.522$ | 9,984 |
| ATD | $85.01 \pm 0.224$ | 7,328 | $\mathbf{93.35 \pm 0.357}$ | 2,688 | $\mathbf{70.26 \pm 0.523}$ | 7,296 | $\mathbf{74.15 \pm 0.431}$ | 9,984 |

*Parenthesis shows the number of training samples. Our improvements are statistically significant with $p < 0.05$ (details in appendix D.7).

datasets have enough training samples, we randomly selected a subset of them) for learning a downstream classifier and use all test data. Each experiment is conducted with five different random seeds and the mean and standard deviations are reported. The metrics are the *accuracy* and the *number of learnable parameters*. All models have 32-dim features in the end, except that for two self-supervised baselines and autoencoder variants, which have 128-dim options.

### 4.2.1 Better Classification Accuracy with Fewer Parameters

From Table 2, `ATD` shows comparable or better performance over the baselines. We have also reported the running time per epoch/sweep in Appendix C.7 for all models. Compared to the variant $ATD_{ss-}$, our `ATD` can improve the accuracy by $1.0\% \sim 1.9\%$, which shows the benefit of the inclusion of self-supervised loss. SALS and $ATD_{ss-}$ have similar performance, while their objectives differ in that $ATD_{ss-}$ considers the Frobenius norm of the augmented data. Thus, their accuracy gap is caused by the use of data augmentation. Also, the experiments show that the *fitness* and *alignment* principles are both important. We observe that with a self-supervised loss (i.e., *alignment*), $AE_{ss}$ can give significant improvements over AE, while `ATD` shows $\sim 8\%$ accuracy gain over the self-supervised models on MGH dataset, since we can better preserve the data with a reconstruction loss (i.e., *fitness*).

Moreover, the table shows that tensor based models require fewer parameters, i.e., less than $5\%$ of parameters compared to deep learning models. On HAR, the deep unsupervised models show poor performance due to (i) they may not optimize a large number of parameters on middle-scale dataset; (ii) movement signals in HAR might have few degrees of freedom, which matches well with the low-rank assumption of tensor methods. On large-scale Sleep-EDF, self-supervised models outperforms `ATD` marginally since they have more parameters thus can capture more information.

### 4.2.2 Better Performance in Low-label Rate Scenarios

On the MGH dataset, we also show the effect of varying the amount of training data in Figure 2. We include an end-to-end convolutional neural network (CNN) model based on (Biswal et al., 2018), called *Reference CNN*, which is a supervised model and only uses the training and test sets. To be more readable, we separate the comparison figure into two sub-figures: the left compares our `ATD` model with self-supervised and auto-encoder baselines and the right one compares `ATD` with tensor baselines and the reference model, and the scale of y-axis on two sub-figures are the same.

We find that all unsupervised models outperform the supervised reference CNN model in scenarios with fewer training samples. With more training data, the performance of all models get improved, especially the reference CNN model, which can optimize the encoder and predictive layers in an end-to-end way and finally outperforms our `ATD` when more training samples is available.

### 4.2.3 Stable Results with Hyperparameter Variation

For a comprehensive evaluation, we also conduct ablation studies on the effect of the data augmentation methods and on hyperparameters. Due to space limitation, we move the experimental settings and

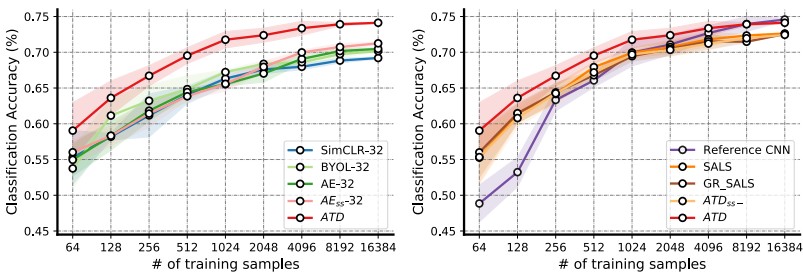

Figure 2: Varying the # of Training Data

results to Appendix C.6, while summarizing the general conclusions here: (i) with more diverse data augmentation methods, the final results are relatively better; (ii) with a larger rank $R$, the performance will be better generally; (iii) our ATD is not sensitive hyperparameters $\alpha$ and $\gamma$, and $\beta \neq 0$ can be chosen from a large range (e.g., $\beta = 2$ in the experiments) for decent performances.

## 5  Related Work

**Data augmentation and Self-supervised Learning.** Data augmentation exploits class-preserving perturbations to smooth out noise and encode task-invariances (Dao et al., 2019). It has been widely used in various data formats, such as images (Cireşan et al., 2010), text (Lu et al., 2006), audio (Uhlich et al., 2017), and time series (Wen et al., 2020; Yang et al., 2021b). Data augmentation also benefits the recent development of contrastive SSL (He et al., 2020; Chen et al., 2020), which extracts class-relevant features by optimizing a deep neural network encoder to achieve agreements between semantically similar samples and disagreements on dissimilar samples. However, in contrastive SSL, a recent work (Arora et al., 2019) highlighted that the common practice of replacing negative samples with random samples leads to sampling bias, which may hurt the learned representation significantly (Chuang et al., 2020). In this paper, we introduce an unbiased self-supervised objective into CP tensor decomposition model and shows that the new design can be helpful in producing class-aware outputs.

**Stochastic Algorithms for Tensors.** With the rapid growth in data volume, efficient stochastic tensor methods become increasingly important for higher-order data structures to boost scalability. These methods are largely based on sampling (Ma and Solomonik, 2021; Yang et al., 2021a; Kolda and Hong, 2020), which accelerates the computation of over-determined least square problems (Battaglino et al., 2018; Larsen and Kolda, 2020) in ALS for dense (Ailon and Chazelle, 2006) and sparse (Eshragh et al., 2019) tensors by effective strategies, such as Fast Johnson-Lindenstrauss Transform (Ailon and Chazelle, 2006), leverage-based sampling (Eshragh et al., 2019), and sketching. However, these algorithms only focus on making ALS steps less costly and require to load the full data into memory. Thus, we do not consider them in our setting. This paper integrates augmentation techniques and self-supervised loss into tensor decomposition, and later we adopt an effective stochastic alternating optimization to handle large scale optimization with less memory consumption.

## 6  Conclusion

This paper introduces the concept of self-supervised learning for tensors and proposes *Augmented Tensor Decomposition* (ATD) and the ALS-based optimization. We show that by explicitly contrasting positive and negative samples, the decomposition results are more aligned with downstream classification. On four real-world datasets, we show the advantages of our model over various unsupervised models and in low-label rate scenarios, our model even outperforms the reference supervised models.

Compared to deep learning methods, tensor based models are not as flexible in processing multimodal and diverse inputs, such as natural images. However, applying tensor decomposition on the outputs of earlier layers of pre-trained deep neural networks may be a feasible way to address the weaknesses. This direction would be interesting for future work.

## Acknowledgements

This work was supported by NSF award SCH-2205289, SCH-2014438, IIS-1838042, NIH award R01 1R01NS107291-01.

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
