# OpenReview forum: "ATD: Augmenting CP Tensor Decomposition by Self Supervision"
_NeurIPS.cc/2022/Conference — NeurIPS 2022 Accept_

### Official Review · Reviewer_qLDm · 2022-07-05

**Rating:** 5
**Confidence:** 4
**Soundness:** 3 good
**Presentation:** 3 good
**Contribution:** 2 fair

**Summary:**

In the paper, the authors investigated a new variant of CP decomposition (CPD) for the classification task. Compared to the classic CPD, the new one imposes the self-supervised loss, so better embedding spaces, modeled by latent factors, can be learned by augmenting the training data.

**Questions:**

1. What is the running time for training the all methods implemented in the experiment?
2. What happens if the augmented data get unreliable, e.g., low-quality?
3. How other tensor methods are influenced by varying the rank in the experiment?

**Limitations:**

The limitation compared with deep learning methods are briefly mentioned, but there is no discussion compared with other tensor-based methods. For example, does the additional loss reduces the interpretability or the expressive power of tensor models?

**Strengths And Weaknesses:**

**Strength:**

- (clarity) The paper is well written. The background, formulation, and algorithm are clearly introduced.
- (originality) Considering the self-supervision loss is new for tensor decomposition, and the experimental results verified the effectiveness of the method.

**Weakness:**

- (quality, clarity) Part of the technical details are not well discussed
    1. Although Thm. 1 proved the convergence for the sub-problem of learning $\mathbf{X}$, only *one* iteration is employed in the whole algorithm as given in the appendix (page 14). It might work well in practice (mentioned in the appendix), but such modification makes me suspect if the whole algorithm enables to converge to fixed points. Careful discussions lack on this point.
    2. It also lacks the discussion (apology if I miss it) regarding how the data augmentation quality impacts the new method's performance.
    3. In the ablation study, the comparison with other tensor baselines should be added. Otherwise, it is hard to fairly evaluate if the proposed method is non-sensitive to the parameters, e.g., the CP rank.
    4. The proposed method's advantage of the number of learnable parameters is not clearly discussed. For example, the deep learning models may run faster than tensor-based ones on GPU even though more parameters are required. If the aim is to evaluate the expressive power of the model, the number of parameters also seems not a good metric, especially when the model architectures are different.
- (significance) The proposed method might be useful for specific tasks such as those in the experiment section. But there are few new ideas or contributions to the tensor community.

---

> ### Author Response · Authors · 2022-07-31
> **Response to Reviewer qLDm (Part I)**
>
> We thank the reviewer for the helpful feedback. We have uploaded a revision and used blue color to mark the new changes. Our detailed responses are as follows.
>
> **Q1. If the whole algorithm enables to converge to fixed points with one iteration. Careful discussions lack on this point.**
>
> Thanks, we have added further discussions to appendix B. Please Check Figure 3, which proves the linear convergence and shows us that with very few iterations (such as two), the precision of the "iterative rule" can already hit the limit of floating-value calculation in our machine. During the rebuttal, we added the results of Sleep-EDF here, which are similar to what we have shown on HAR data. We could have two iterations instead of one, however, as shown in Table 3 and Table 4 of the revision, one iteration already gives us good accuracy with less time consumption. In practice, the iteration number can increase per application need and precision requirements.
>
> **Q2. How the data augmentation quality impacts the new method's performance?**
>
> Thanks, this is a good question. During the rebuttal, we tried different experimental scenarios with low-quality augmentation methods:
>
> - We create low-quality **jittering** methods with larger $d$, and $d$ is the amplitude of the added noise. For reference, $d=1$ means the amplitude of the noise is the same as the signal. In the experiments of Table 2, we use $d=0.01$ to construct good-quality augmentation for MGH data.
> - Scenario (A): we use a bad **jittering** method and the good **bandpass filtering** method.
> - Scenario (B): we use the good **jittering** method and a bad **bandpass filtering** method.
> - Scenario (C): we use a bad **jittering** method and a bad **bandpass filtering** method.
>
> Details of these scenario settings can be found in appendix D.5. We summarize the results below,
>
> | MGH | d=0.02 | d=0.05   | d=0.1 | d=0.5 | d=5|
> |-----------|:-----------:|:-------:|:--------:|:-------:|:-----:|
> |Accuracy (%) | 74.18 ± 0.326 | 74.10 ± 0.302 | 73.85 ± 0.530 | 73.39 ± 0.493 | 72.18 ± 0.676 |
>
> | MGH | SALS | ATD   | Scenario (A) | Scenario (B) | Scenario (C)|
> |-----------|:-----------:|:-------:|:--------:|:-------:|:-------:|
> |Accuracy (%) | 71.93 ± 0.379 | **74.15 ± 0.431** | 72.73 ± 0.624 | 72.10 ± 0.719 | 70.75 ± 0.771|
>
> | HAR | SALS | ATD   | Scenario (A) | Scenario (B) | Scenario (C)|
> |-----------|:-----------:|:-------:|:--------:|:-------:|:-------:|
> |Accuracy (%) | 91.86 ± 0.295 | **93.35 ± 0.357** | 92.04 ± 0.308 | 92.48 ± 0.469 | 91.43 ± 0.835|
>
> Based on these new results, the conclusion is that low-quality data augmentations will negatively impact the method’s performance. However, the performance degradation is empirically small. The reason is that in this case, the Frobenius fitness loss might dominate and the model degrades gradually to SALS model with extra noise (introduced by low-quality data augmentation and the self-supervised loss in this case).
>
> **Q3. How other tensor methods are influenced by varying the rank in the experiment?**
>
> Thanks for the suggestions. During the rebuttal, we compared with other tensor methods under different R values and show the results below (more details in appendix D.6). The conclusion is that (i) with a larger R, all tensor models tend to output better results while the model performance does not change much beyond R=32; (ii) our model ATD consistently works better compared to other tensor baselines.
>
> - **On HAR dataset**
>
> | Model (%) | R=8 | R=16   | R=32 | R=64 | R=128|
> |-----------|:-----------:|:-------:|:--------:|:-------:|:-------:|
> | SALS | 69.59 ± 0.526 | 83.92 ± 0.416 | 91.84 ± 0.295|91.89 ± 0.217| 91.55 ± 0.388|
> |GR-SALS|69.62 ± 0.458|84.20 ± 0.727| 92.33 ± 0.282| 92.28 ± 0.359| 91.84 ± 0.534|
> |$ATD_{ss-}$|70.27 ± 0.488| 84.84 ± 0.557| 92.41 ± 0.391| 92.71 ± 0.243| 92.32 ± 0.330|
> |ATD| **71.91 ± 0.253**| **85.61 ± 0.294**|**93.35 ± 0.357**| **93.43 ± 0.411**| **92.97 ± 0.273**|
>
> > In the case of $R=128$, the logistic regression model might overfit, so the performance of all tensor methods becomes slightly worse.
>
> - **On Sleep-EDF dataset**
>
> | Model (%) | R=8 | R=16   | R=32 | R=64 | R=128|
> |-----------|:-----------:|:-------:|:--------:|:-------:|:-------:|
> | SALS | 81.26 ± 0.345 | 82.59 ± 0.638 | 84.27 ± 0.481|84.55 ± 0.527| 84.49 ± 0.317|
> |GR-SALS|81.72 ± 0.664|82.74 ± 0.481| 84.33 ± 0.356| 84.87 ± 0.486| 84.90 ± 0.781|
> |$ATD_{ss-}$|81.27 ± 0.568| 82.50 ± 0.674| 84.19 ± 0.221| 84.47 ± 0.258| 84.44 ± 0.577|
> |ATD| **82.49 ± 0.464**| **83.31 ± 0.591**|**85.01 ± 0.224**| **85.30 ± 0.483**| **85.32 ± 0.305**|

---

> ### Author Response · Authors · 2022-07-31
> **Response to Reviewer qLDm (Part II)**
>
> **Q4. The advantage of the number of learnable parameters is not clearly discussed. For example, the deep learning models may run faster than tensor-based ones on GPU even though more parameters are required.**
>
> Tensor-based models have some advantages over deep learning models:
>
> - (i) generally, on applications with smaller datasets, over-parameterized models might fail to optimize.
> - (ii) tensor models usually have much fewer parameters compared to deep learning models, such that they are light to store and easy to deploy.
>
> Also, tensor methods actually run faster than deep learning models on GPU, since matrix/tensor operation can be greatly accelerated. We will show that in the next question.
>
> **Q5. What is the running time for training all the methods implemented in the experiment?**
>
> Regarding running time comparison, we added the per-epoch running time for all compared models below (more details in appendix D.7). All models run on GPUs.
>
> | Model (%) | Sleep-EDF | HAR   | PTB-XL | MGH   |
> |-----------|:-----------:|:-------:|:--------:|:-------:|
> |SimCLR-32|260.299s|8.459s|18.714s|1449.368s|
> |SimCLR-128|265.809s|8.532s|19.037s|1457.283s|
> |BYOL-32|255.614s|8.430s|18.410s|1451.468s|
> |BYOL-128|257.266s|8.478s|18.680s|1461.181s|
> |AE-32|153.684s|7.530s|11.229s|851.118s|
> |AE-128|156.813s|7.662s|11.396s|815.858s|
> |$AE_{ss}$-32|301.773s|7.765s|18.263s|1486.244s|
> |$AE_{ss}$-128|307.546s|7.804s|18.465s|1504.545s|
> |SALS|86.281s|7.535s|8.988s|782.763s|
> |GR_SALS|109.916s|7.829s|9.747s|970.292s|
> |$ATD_{ss-}$|147.568s|8.604s|12.560s|1327.188s|
> |ATD|148.375s|8.672s|12.599s|1360.569s|
>
> **Q6. The proposed method might be useful for specific tasks such as those in the experiment section. But there are few new ideas or contributions to the tensor community.**
>
> Thanks. We can list some future directions and application scenarios that can be inspired or benefit from our method:
> - First, we introduced two useful techniques "data augmentation" and "self-supervised learning" from the deep learning area to tensor decomposition and use extensive experiments to demonstrate that they can improve the classification task in tensors. This "Siamese"-type tensor framework (shown in Figure 1) can inspire several follow-up works, such as using "good-quality guided tensors" in the augmented part to guide the decomposition of a target tensor with some customized objectives/criteria.
> - Second, this paper proposed a new unsupervised way of learning better subspaces, so as to extract more accurate low-dimensional representations from tensor data. Imagine the scenario where a domain expert wants to perform tensor decomposition on a small dataset, she/he can transform the domain knowledge into useful data augmentations and apply our methods to reduce noise and obtain better decomposition results.
> - Third, although we build our model on common CP decomposition in the paper and use float-value-based inputs for experiments. Our main idea can possibly be extended to Tucker decomposition and bounded/categorical-valued tensor inputs.
>
> **Q7. Discussion of limitations compared with other tensor-based methods**
>
> Tensor-based methods are all about learning good subspaces/bases (at least in our experimental setting). Thus, the additional loss will not reduce the interpretability or the expressive power of our model. Instead, with the (class-preserving) data augmentation methods and self-supervised loss, our model learns better and more robust subspaces than the baseline models, such that we achieve better downstream classification performance.
>
> Regarding potential limitations of our model:
> - First, by using data augmentations, the batch size of our model doubles (original tensor plus the augmented tensor). Thus, the running time of our model would be greater than the common mini-batch methods (e.g., SALS), which might be a limitation. However, more training data means faster convergence. We empirically find that ATD only needs a half number of the epochs to converge, compared to SALS.
> - Second, as discussed before, poor-quality data augmentations might hurt the performance, so how to choose/design (or even automatically generate) better augmentation techniques can be future work.
>
> In the end, we sincerely thank the reviewer again for the constructive comments. Hope our new results and explanations can clear all your concerns. We are open to further discussions on these details.

---

### Official Review · Reviewer_8z4A · 2022-07-06

**Rating:** 6
**Confidence:** 2
**Soundness:** 3 good
**Presentation:** 3 good
**Contribution:** 3 good

**Summary:**

This paper propose a novel self-supervised framework for CP Tensor Decomposition, in which an iterative method is developed to enable  an alternative least squares fashion. The experiment results on various datasets show its efficiency.

**Questions:**

 could the authors provide more explaination about the derivation of Eq.(8)?

**Ethics Review Area:**

["I don’t know"]

**Strengths And Weaknesses:**


1. incorporating self-supervised learning into CP Tensor Decomposition seems to new to me.
2. the experiments are extensive and promising
3. this paper is well-motivated

---

> ### Author Response · Authors · 2022-07-27
> **Response to Reviewer 8z4A**
>
> Yes, we can provide the derivations of Eq. (8) below. This derivation is also added to appendix E in the revision.
>
> ---
> ### **Statement of the two-sided bound in Eq. (8):**
>
> We recall the definition of $L_{ss}$ and $L^\Theta_{ss}$ from Eq. (6) and Eq. (7).
>
> > $$L_{ss} = L_{pos} + \lambda L_{neg}= E \left[\frac{\lambda}{1-r_p}sim\left(f\left(X_p \right),f\left({Y_q}\right)\right)\right] - E\left[\left(\frac{\lambda r_p}{1-r_p}+1\right)sim\left(f\left(X_p \right),f\left({Y_q}\right)\right) \mid p = q \right], $$
>
> > $$
> L^\Theta_{ss}(\gamma) =(\gamma+1)E \left[sim\left(f\left(X_p\right),f\left({Y_q}\right)\right)\right] -E \left[sim\left(f\left(X_p\right),f\left({Y_q}\right)\right)\mid p= q\right],
> $$
>
> and we want to prove that
>
> $$
> \begin{equation}
> 	C_1L^\Theta_{ss}\left(\frac{\lambda-1}{C_1}\right) \leq L_{ss} \leq C_2L^\Theta_{ss}\left(\frac{\lambda-1}{C_2}\right), C_1=1+\max_p\frac{\lambda r_p}{1-r_p}, C_2=1+\min_p\frac{\lambda r_p}{1-r_p},
> \end{equation}
> $$
>
> where $r_p$ is the label rate of class-$p$.
>
> ---
> ###  **Derivation of the two-sided bound:**
>
> We start by arranging $L_{ss}$,
>
> $$L_{ss} =   E \left[\frac{\lambda}{1-r_p} sim\left(f\left(X_p \right),f\left({Y_q}\right)\right)\right] -   E\left[\left(\frac{\lambda r_p}{1-r_p}+1\right) sim\left(f\left(X_p \right),f\left({Y_q}\right)\right) \mid p = q \right] \notag$$
> 	$$=   E \left[\left(\frac{\lambda r_p}{1-r_p} + \lambda\right) sim\left(f\left(X_p \right),f\left({Y_q}\right)\right)\right] -   E\left[\left(\frac{\lambda r_p}{1-r_p}+1\right) sim\left(f\left(X_p \right),f\left({Y_q}\right)\right) \mid p = q \right]\notag $$
> 	$$=   E \left[\left(\frac{\lambda r_p}{1-r_p} + 1\right) sim\left(f\left(X_p \right),f\left({Y_q}\right)\right)\right]
> 	+   E \left[\left(\lambda- 1\right) sim\left(f\left(X_p \right),f\left({Y_q}\right)\right)\right] \notag-   E\left[\left(\frac{\lambda r_p}{1-r_p}+1\right) sim\left(f\left(X_p \right),f\left({Y_q}\right)\right) \mid p = q \right]$$
>     $$=   E_p  E_{q,X_p,Y_q} \left[\left(\frac{\lambda r_p}{1-r_p} + 1\right) sim\left(f\left(X_p \right),f\left({Y_q}\right)\right)\right]
> 	+   E \left[\left(\lambda- 1\right) sim\left(f\left(X_p \right),f\left({Y_q}\right)\right)\right] -   E_p  E_{q,X_p,Y_q}\left[\left(\frac{\lambda r_p}{1-r_p}+1\right) sim\left(f\left(X_p \right),f\left({Y_q}\right)\right) \mid p = q \right] $$
> 	$$=   E_p\left[\left(\frac{\lambda r_p}{1-r_p} + 1\right)  E_{q,X_p,Y_q} \left[ sim\left(f\left(X_p \right),f\left({Y_q}\right)\right)\right]\right]
> 	+   E \left[\left(\lambda- 1\right) sim\left(f\left(X_p \right),f\left({Y_q}\right)\right)\right] -   E_p\left[\left(\frac{\lambda r_p}{1-r_p}+1\right)  E_{q,X_p,Y_q}\left[ sim\left(f\left(X_p \right),f\left({Y_q}\right)\right) \mid p = q \right]\right]$$
> 	$$=   E_p\left[\left(\frac{\lambda r_p}{1-r_p} + 1\right)\left(  E_{q,X_p,Y_q} \left[ sim\left(f\left(X_p \right),f\left({Y_q}\right)\right)\right]-  E_{q,X_p,Y_q}\left[ sim\left(f\left(X_p \right),f\left({Y_q}\right)\right) \mid p = q \right]\right)\right] +   E \left[\left(\lambda- 1\right) sim\left(f\left(X_p \right),f\left({Y_q}\right)\right)\right]$$
> 	$$\leq   E_p\left[C_2\left(  E_{q,X_p,Y_q} \left[ sim\left(f\left(X_p \right),f\left({Y_q}\right)\right)\right]-  E_{q,X_p,Y_q}\left[ sim\left(f\left(X_p \right),f\left({Y_q}\right)\right) \mid p = q \right]\right)\right] +   E \left[\left(\lambda- 1\right) sim\left(f\left(X_p \right),f\left({Y_q}\right)\right)\right] $$
> 	$$= (C_2+\lambda-1)  E \left[ sim\left(f\left(X_p \right),f\left({Y_q}\right)\right)\right]-C_2  E\left[ sim\left(f\left(X_p \right),f\left({Y_q}\right)\right) \mid p = q \right] \notag$$
> 	$$= C_2L^{\Theta}_{ss}\left(\frac{\lambda-1}{C_2}\right).$$
>
>
> In this derivation, we use
> - **Fact 1**: $E[\cdot]$ indicates that the expectation is taken over four interdependent random variables, i.e., $p,q,X_p,Y_q$, which is defined in Section 3.1.
> - **Fact 2**: Given $p$, the similarity of random pairs is smaller than the similarity of positive pairs $E_{q,X_p,Y_q} \left[sim\left(f\left(X_p\right),f\left({Y_q}\right)\right)\right]\leq E_{q,X_p,Y_q}\left[sim\left(f\left(X_p \right),f\left({Y_q}\right)\right) \mid p = q \right]$.
>
> The upper bound is derived by replacing $\frac{\lambda r_p}{1-r_p} + 1,~\forall p$ with $C_2=1+\min_{p}\frac{\lambda r_p}{1-r_p}$. Similarly, we can also derive the other side (lower bound) by using $C_1=1+\max_{p}\frac{\lambda r_p}{1-r_p}$, which eventually gives $C_1L^\Theta_{ss}\left(\frac{\lambda-1}{C_1}\right)$. The equivalence of two bounds is established when $C_1 = C_2$, i.e., the class labels are balanced.
>
> ---
> Thanks for your question, hope our explanation has answered it. We are also open to having more technical discussions.

---

### Official Review · Reviewer_cxKw · 2022-07-09

**Rating:** 8
**Confidence:** 4
**Soundness:** 4 excellent
**Presentation:** 4 excellent
**Contribution:** 3 good

**Summary:**

The authors present  "augmented" tensor decomposition, which utilizes an objective that is designed for downstream tasks (by way of "alignment") in addition to data reconstruction ("fitness"). Hence, the quality of the decomposition is determined not only by reconstruction error, but by correct clustering on features. The authors contribute an algorithm for implementing this approach and motivate the significance with a comparison to existing CP decomposition approaches. The authors' algorithm builds on Alternating Least Squares (ALS), a typical method in CP decomposition algorithms.

**Questions:**

- Is there a concern about the validity of using classification accuracy to measure model performance, in cases where classes are highly imbalanced?
- Appendix D.6 addresses that ATD's performance improves with larger R. What about its performance relative to the other models in Table 2?


**Limitations:**

- Good discussion of limitations. Please emphasize whether the computational complexity compared to existing methods is a limitation or strength. One obvious limitation (given that medical data is heavily used in the experiments on real data) that may be useful to mention is that this method is developed for real-valued data (as opposed to bounded data).

**Strengths And Weaknesses:**

Strengths:
- The approach is well motivated. The novelty and potential significance of the authors' contributions is made immediately clear in Section 1.
- The contributions are clearly defined and include: a method to generate dissimilar data samples in the absence of labels.
- Presentation is clear and easy to follow.
- The experiments on real data use a convincing variety of datasets: there is sufficient variation in dataset size, balance of dimensions, and downstream tasks.

Weaknesses:
- The discussion of classification appears to be wholly motivated by classification of samples using features (eg. classifying patients based on EEG-related features). The nature of the classification task is heavily dependent on the application -- eg. decomposition of DNA methylation data (where it is desirable to do dimensionality reduction on the # of features, which is very large, in addition to clustering patients) is often motivated by simultaneous clustering/classification of features in addition to patients. Would be helpful to clarify whether more flexible classification is feasible/reasonable under this approach.
- The perennial question in CP/Tucker decomposition is how to choose the dimension of latent features (ie. the decomposition rank). Would like to see more discussion on this topic, eg. the stability of this method's performance to perturbations in this regard (ex. is ATD still superior if R is varied within a reasonable range?)
- The datasets in Section 4 (Experiments) all have a relatively similar # of classes. The results would be more compelling if there was more variation in this dimension.

---

> ### Author Response · Authors · 2022-07-31
> **Response to Reviewer cxKw (Part I)**
>
> We thank the reviewer for the appreciation of our paper and the helpful comments. We have uploaded a revision and used blue color to mark the new changes. Our detailed responses are as follows:
>
> **Q1. Would be helpful to clarify whether more flexible classification is feasible/reasonable under this approach.**
>
> This is a good suggestion. Our model can be applied as long as the date input admits a component-based structure. More flexible classification can be realized by adding meaningful customized objectives to Eq (10) and the final optimization procedures might change accordingly. For example, we can add the K-means loss function (using X and X’ as variables), which is convex as well. Then, the optimization method can be nested alternating least squares (ALS) and EM algorithm (E step: estimate the cluster assignment; M step: run ALS to optimize A, B, C, X, X’ and the cluster centers).
>
> **Q2. How to choose the dimension of latent features $R$? Is ATD still superior if $R$ is varied within a reasonable range.**
>
> Thanks for the question. The choice of $R$ depends on the trade-off between model fitness and time complexity. Specifically, a larger $R$ means better fitness and more preserved information in the extracted representations, while the number of learnable parameters and time complexity also increases linearly with $R$. In our paper, we run simple CP decomposition on a small subset of the tensor and monitor the fitness curve. We find that the fitness does not improve much around $R=32$ for all datasets (which means the real tensor might have a smaller rank and the residual part might be just noise). Thus, we choose $R=32$ throughout the paper.
>
> We have conducted ablation study on other tensor-based methods with $R=8, 16, 32, 64, 128$ and show the results below on HAR and Sleep-EDF datasets (details in appendix D.6). The conclusions are that (i) with a larger $R$, all models tend to give better performance while the performance does not change much beyond $R=32$; (ii) our model ATD consistently works better than the tensor baselines.
>
> - **On HAR dataset**
>
> | Model (%) | R=8 | R=16   | R=32 | R=64 | R=128|
> |-----------|:-----------:|:-------:|:--------:|:-------:|:-------:|
> | SALS | 69.59 ± 0.526 | 83.92 ± 0.416 | 91.84 ± 0.295|91.89 ± 0.217| 91.55 ± 0.388|
> |GR-SALS|69.62 ± 0.458|84.20 ± 0.727| 92.33 ± 0.282| 92.28 ± 0.359| 91.84 ± 0.534|
> |$ATD_{ss-}$|70.27 ± 0.488| 84.84 ± 0.557| 92.41 ± 0.391| 92.71 ± 0.243| 92.32 ± 0.330|
> |ATD| **71.91 ± 0.253**| **85.61 ± 0.294**|**93.35 ± 0.357**| **93.43 ± 0.411**| **92.97 ± 0.273**|
>
>
> > In the case of $R=128$, the logistic regression model might overfit, so the performance of all tensor methods becomes slightly worse.
>
> - **On Sleep-EDF dataset**
>
> | Model (%) | R=8 | R=16   | R=32 | R=64 | R=128|
> |-----------|:-----------:|:-------:|:--------:|:-------:|:-------:|
> | SALS | 81.26 ± 0.345 | 82.59 ± 0.638 | 84.27 ± 0.481|84.55 ± 0.527| 84.49 ± 0.317|
> |GR-SALS|81.72 ± 0.664|82.74 ± 0.481| 84.33 ± 0.356| 84.87 ± 0.486| 84.90 ± 0.781|
> |$ATD_{ss-}$|81.27 ± 0.568| 82.50 ± 0.674| 84.19 ± 0.221| 84.47 ± 0.258| 84.44 ± 0.577|
> |ATD| **82.49 ± 0.464**| **83.31 ± 0.591**|**85.01 ± 0.224**| **85.30 ± 0.483**| **85.32 ± 0.305**|
>
> **Q3. The datasets in Section 4 all have a relatively similar # of classes. The results would be more compelling if there was more variation in this dimension.**
>
> Thanks for the suggestion. We have already included four datasets from different applications with up to six classes. Due to time limitations, we are unable to show additional results at this time. Will consider adding new datasets with more classes (such as Fashion-MNIST) after the rebuttal.
>
> **Q4. Is there a concern about the validity of using classification accuracy to measure model performance, in cases where classes are highly imbalanced?**
>
> Thanks for raising the concern. In cases with highly imbalanced datasets, we will consider using (weighted) AUROC or F1 metrics. However, handling imbalanced data is beyond the main scope of this paper, since the datasets we used are mostly balanced (statistics are shown in Table 5).

---

> > ### Comment · Reviewer_cxKw · 2022-08-07
> > **Rebuttal received**
> >
> > I thank the authors for the thorough, thoughtful response to my initial review, and their revisions to the submission. I maintain my initial rating to recommend acceptance.

---

> ### Author Response · Authors · 2022-07-31
> **Response to Reviewer cxKw (Part II)**
>
> **Q5. Please emphasize whether the computational complexity compared to existing methods is a limitation or a strength.**
>
> Sure, we would like to clarify this. We have analyzed the complexity at the end of Section 3.3, and our ATD has the same asymptotical complexity as common CP-ALS, which is $O(NIJKR)$ and $N\times I\times J\times K$ is the input tensor size. In fact, all our tensor-based baselines have the same asymptotical complexity. Compared to deep learning models, tensor methods are much more efficient. To prove it, we add the per epoch running time below. Since our model ATD and the variant $ATD_{ss−}$ use the augmented tensors (so training size doubles), they cost more compared to other tensor-based methods while empirically they require a half number of the epochs to converge.
>
> | Model (%) | Sleep-EDF | HAR   | PTB-XL | MGH   |
> |-----------|:-----------:|:-------:|:--------:|:-------:|
> |SimCLR-32|260.299s|8.459s|18.714s|1449.368s|
> |SimCLR-128|265.809s|8.532s|19.037s|1457.283s|
> |BYOL-32|255.614s|8.430s|18.410s|1451.468s|
> |BYOL-128|257.266s|8.478s|18.680s|1461.181s|
> |AE-32|153.684s|7.530s|11.229s|851.118s|
> |AE-128|156.813s|7.662s|11.396s|815.858s|
> |$AE_{ss}$-32|301.773s|7.765s|18.263s|1486.244s|
> |$AE_{ss}$-128|307.546s|7.804s|18.465s|1504.545s|
> |SALS|86.281s|7.535s|8.988s|782.763s|
> |GR_SALS|109.916s|7.829s|9.747s|970.292s|
> |$ATD_{ss-}$|147.568s|8.604s|12.560s|1327.188s|
> |ATD|148.375s|8.672s|12.599s|1360.569s|
>
> **Q6. One obvious limitation that may be useful to mention is that this method is developed for real-valued data (as opposed to bounded data).**
>
> Thanks for the suggestion. In this paper, we build our model on common CP decomposition and use float-value-based inputs for experiments. However, our model may apply to other tensor models that take bounded data as inputs, such as boolean tensor decomposition [1], which could extend our model to bounded data applications.
>
> [1] Miettinen, P. (2011, December). Boolean tensor factorizations. In 2011 IEEE 11th International Conference on Data Mining (pp. 447-456). IEEE.
>
> ---
> We thank the reviewer again for the detailed and insightful comments. Hope our additional results and the rebuttal have cleared all the concerns. We are happy to have further discussions on these technical details.

---

### Official Review · Reviewer_rtia · 2022-07-10

**Rating:** 5
**Confidence:** 4
**Soundness:** 3 good
**Presentation:** 4 excellent
**Contribution:** 3 good

**Summary:**

This paper proposes a tensor decomposition leveraging contrastive self-supervised learning based on the data augmentation for positive samples and the unbiased estimation of negative samples, which can align with downstream classification tasks. The proposed method optimizes the objective by using an alternating least squares algorithm. The experimental results using four real-world tensor datasets show that the proposed method achieves better accuracy on the downstream classification tasks using a much smaller number of parameters than several comparison models including tensor-based models, self-supervised models, and auto-encoder models.

**Questions:**

Have you compared the proposed method with some supervised tensor learning by using the training/test dataset?

**Limitations:**

As mentioned above, the classification accuracy is not significantly different with or without the empirical self-supervised loss in the experimental results. It should be made more clear the situation the proposed method is strongly needed.

**Strengths And Weaknesses:**

Originality:
The proposition of contrastive learning using data augmentation for positive samples will greatly contribute to the research community of tensor decomposition. Although the proposed method is a fundamentally unsupervised method and able to utilize a large number of unlabeled data, it should be compared with several works of supervised tensor learning such as [Tao et al., 2005].

[Tao et al., 2005] Dacheng Tao, Xuelong Li, Weiming Hu, Stephen J. Maybank, Xindong Wu: Supervised Tensor Learning. ICDM 2005: 450-457

Quality:
The problem setting makes sense. The proposed method is convincing along with theoretical discussion.

Clarity:
This paper is well-organized and clearly written.

Significance:
The classification accuracy shown in Table 2 is not significantly different with or without the empirical self-supervised loss. Moreover, as described in the appendix, the proposed method is not sensitive to beta. There is room to make the proposed method more significant.

---

> ### Author Response · Authors · 2022-07-31
> **Response to Reviewer rtia**
>
> We thank the reviewer for the helpful feedback. We have uploaded a revision with the changes marked as blue. Our detailed responses are as follows:
>
> **Q1. Although the proposed method is a fundamentally unsupervised method and able to utilize a large number of unlabeled data, it should be compared with several works of supervised tensor learning.**
>
> Thanks for your suggestion. We added two supervised tensor learning baselines UMLDA [1] and STL [2] and have shown the performance below. Our model significantly outperforms two supervised tensor learning baselines. Details can be founded in appendix D.8 of the revision.
>
> [1] Lu, H., Plataniotis, K. N., & Venetsanopoulos, A. N. (2008). Uncorrelated multilinear discriminant analysis with regularization and aggregation for tensor object recognition. *IEEE Transactions on Neural Networks*, *20* (1), 103-123.
>
> [2] Tao, D., Li, X., Hu, W., Maybank, S., & Wu, X. (2005, November). Supervised tensor learning. In *Fifth IEEE International Conference on Data Mining (ICDM'05)* (pp. 8-pp). IEEE.
>
> | Model (%) | Sleep-EDF | HAR   | PTB-XL | MGH   |
> |-----------|:-----------:|:-------:|:--------:|:-------:|
> | UMLDA     | 81.06 ± 0.093	| 85.73 ± 1.169	| 65.55 ± 0.267	|62.04 ± 0.722 |
> | STL       | 77.86 ± 0.816	| 80.52 ± 0.189 |	61.83 ± 0.712 |	41.44 ± 0.597 |
> | Our ATD   | **85.01 ± 0.224**|	**93.35 ± 0.357**|	**70.26 ± 0.523**	|**74.15 ± 0.431**|
>
> **Q2. The classification accuracy shown in Table 2 is not significantly different with or without the empirical self-supervised loss.**
>
> - [p-values of the results] — Thanks, we have calculated the p-values of the t-test below (in the parenthesis) based on Table 2. The results show that our model outperforms the best baseline by 0.8%~2.5% accuracy relatively with p-values smaller than 0.0129. Commonly, a 0.05 p-value is considered statistically significant.
>
>     | Model (%) | Sleep-EDF | HAR   | PTB-XL | MGH   |
>     |-----------|:-----------:|:-------:|:--------:|:-------:|
>     | SALS | 84.27 ± 0.481 (**0.0041**) | 91.86 ± 0.295 (**2e-5**) | 69.15 ± 0.483 (**0.0023**) | 71.93 ± 0.379 (**5e-6**) |
>     |GR-SALS | 84.33 ± 0.356 (**0.0019**) | 92.33 ± 0.282 (**0.0003**) | 69.02 ± 0.477 (**0.0012**) | 72.35 ± 0.228 (**8e-6**) |
>     |$ATD_{ss-}$| 84.19 ± 0.221 (**9e-5**) | 92.41 ± 0.391 (**0.0011**) | 69.38 ± 0.612 (**0.0129**) | 72.78 ± 0.522 (**0.0005**) |
>     |ATD| **85.01 ± 0.224** | **93.35 ± 0.357** | **70.26 ± 0.523** | **74.15 ± 0.431**|
>
> - [ATD vs. $ATD_{ss-}$ vs. SALS]— SALS is the mini-batch CPD model and $ATD_{ss-}$ is our model variant without self-supervised loss but it has the augmented tensor. $ATD_{ss-}$ works better than SALS means that the data augmentation methods are effective. ATD with self-supervised loss further improves the performance proving that the self-supervised loss enables the model to learn better subspaces by filtering out class-irrelevant noise.
>
> **Q3. As described in the appendix, the proposed method is not sensitive to beta. There is room to make the proposed method more significant.**
>
> Thanks, we added more clarifications to appendix D.6. Being insensitive to $\beta$ means that it is easy for users to select a $\beta\neq0$ for decent performance. In Figure 5, we can see that on Sleep-EDF dataset, selecting a $\beta\in[5, 125]$ would guarantee good results while on HAR, the selection range is like $[5, 25]$. The choice of $\beta$ will affect the final performance, but it is not related to the significance of the results.
>
> **Q4. It should be made more clear the situation the proposed method is strongly needed.**
>
> Thanks for this suggestion. Our method improves learning in (1) small data and (2) low resource settings.
>
> - As shown in Figure 2, our model can largely improve the learning from small datasets, where the available training data is not enough to learn good tensor subspaces. In this case, the extra knowledge and variations given by data augmentations and self-supervised loss would be more beneficial for learning class-invariant features and preventing overfitting.
> - For low resource settings, such as smaller systems with limited storage and memory, our method would be preferred, since it requires much fewer learnable parameters and the optimized model is also light to store.
>
> Thanks again for the valuable comments. Hope our new results and the rebuttal have addressed all your concerns.

---

> > ### Comment · Reviewer_rtia · 2022-08-10
> > **Reviewer Response**
> >
> > I thank the authors for their detailed response. The authors have solved my concerns by adding the results and explanations.

---

### Meta-Review · Area_Chair_3b75 · 2022-08-27

**Recommendation:** Accept
**Confidence:** Certain

**Metareview:**

The authors proposes augmented tensor decomposition (ATD) to adapt data reconstruction towards the downstream tasks, e.g., appropriate feature clustering.  It leverages data augmentation and self-supervised learning, with optimization accomplished akin to alternating least square (ALS).  Significantly superior performance is obtained in experiment, compared with existing CP methods and ALS.

All the reviewers, including myself, find the paper a solid contribution to the methodology and analysis. There were a few concerns and clarification requests, and the rebuttal did a good job addressing them. These additional results and insights can be included in the final version of the paper.

**Award:**

No

---

### Decision · Program_Chairs · 2022-09-14

Accept